# Adjustable Sound Absorber of Multiple Parallel-Connection Helmholtz Resonators with Tunable Apertures Prepared by Low-Force Stereolithography of Photopolymer Resin

**DOI:** 10.3390/polym14245434

**Published:** 2022-12-12

**Authors:** Fei Yang, Shaohua Bi, Xinmin Shen, Zhizhong Li, Xiangpo Zhang, Enshuai Wang, Xiaocui Yang, Wenqiang Peng, Changchuang Huang, Peng Liang, Guoxin Sun

**Affiliations:** 1Field Engineering College, Army Engineering University of PLA, Nanjing 210007, China; 2State Key Laboratory of Disaster Prevention & Mitigation of Explosion & Impact, College of Defense Engineering, Army Engineering University of PLA, Nanjing 210007, China; 3Engineering Training Center, Nanjing Vocational University of Industry Technology, Nanjing 210023, China; 4MIIT Key Laboratory of Multifunctional Lightweight Materials and Structures (MLMS), Nanjing University of Aeronautics and Astronautics, Nanjing 210016, China; 5College of Aerospace Science and Engineering, National University of Defense Technology, Changsha 410073, China

**Keywords:** adjustable acoustic metamaterials, TA–MPCHR, sound absorption performance, photopolymer resin, low-force stereolithography, acoustic finite element simulation, various noise control

## Abstract

The variable noise spectrum for many actual application scenarios requires a sound absorber to adapt to this variation. An adjustable sound absorber of multiple parallel-connection Helmholtz resonators with tunable apertures (TA–MPCHRs) is prepared by the low-force stereolithography of photopolymer resin, which aims to improve the applicability of the proposed sound absorber for noise with various frequency ranges. The proposed TA–MPCHR metamaterial contains five metamaterial cells. Each metamaterial cell contains nine single Helmholtz resonators. It is treated as a basic structural unit for an array arrangement. The tunable aperture is realized by utilizing four segments of extendable cylindrical chambers with length *l*_0_, which indicates that the length of the aperture *l* is in the range of [*l*_0_, 4*l*_0_], and that it is tunable. With a certain group of specific parameters for the proposed TA–MPCHR, the influence of the tunable aperture with a variable length is investigated by acoustic finite element simulation with a two-dimensional rotational symmetric model. For the given noise spectrum of certain actual equipment with four operating modes, the TA–MPCHR sample with a limited total thickness of 40 mm is optimized, which is made of photopolymer resin by the low-force stereolithography, and its actual average sound absorption coefficients for the frequency ranges of 500–800 Hz, 550–900 Hz, 600–1000 Hz and 700–1150 Hz reach 0.9203, 0.9202, 0.9436 and 0.9561, respectively. Relative to common non-adjustable metamaterials, the TA–MPCHR made of photopolymer resin can reduce occupied space and improve absorption efficiency, which is favorable in promoting its practical applications in the noise pollution prevention.

## 1. Introduction

Noise refers to sound that interferes with the surrounding living environment, such as industrial noise [1], construction noise [2], traffic noise [3] and social activity noise [4]. Normally, the actual noise is in a wide frequency region [5], which requires the utilized sound absorber to be broadband [6,7,8,9,10]. Heo et al. [6] utilized acoustic metasurfaces (AMSes) with high reflective characteristics to achieve significant noise reduction in tire–pavement interactions over a broadband of acoustic frequencies under 1000 Hz. A novel sound absorber of acoustic metamaterial by the parallel connection of multiple spiral chambers that exhibit an extraordinarily low-frequency sound absorption performance in the frequency range of 360–680 Hz has been proposed by Duan et al. [8], which was prepared by the low-force stereolithography of the photopolymer resin. Setyowati et al. [10] studied the sound absorption performance of absorber-based biomass fiber-reinforced polyester resins, and it showed that a sample of a 15 mm single-tailed cavity kenaf fiber had a higher sound absorption and wider broadband frequencies than the hemp fiber, with a peak on 0.31–0.32 between 1.00–2.00 kHz. The achievement of broad low-frequency sound absorption performance promotes the application of acoustic metamaterials.

Meanwhile, in most cases, the spectrum of noise is not usually immutable [11], which demands that applied sound absorbing materials or structures must be adjustable [12,13,14,15,16,17,18]. Xing et al. [12] proposed the adjustable membrane-type acoustic metamaterials with polyethylene terephthalate (PET) for low-frequency sound absorption, which was realized by changing geometric stiffness to adjust the sound absorption peak. The acoustic multi-layer Helmholtz resonance metamaterials with multiple adjustable absorption peaks was developed by Duan et al. [13], which was fabricated by the low-force stereolithography of the photopolymer resin. Additionally, it could achieve two groups of resonance peaks at 100 and 400 Hz with the thickness of the structure only 1/30th of the working wavelength. Zhai et al. [14] proposed a kind of tunable and artificial composite acoustic material made of photopolymer resin by 3D printing technology to obtain perfect sound energy absorption at certain frequencies. Additionally, the relative impedance of the metamaterial could be changed by rotating the inner splitting ring around the axis. A tunable low-frequency acoustic absorber composed of multi-layered ring-shaped microslit resin tubes with deep subwavelength thickness was developed by Xu et al. [15], and highly efficient acoustic absorption was achieved in the range of 280–572 Hz. Jiang et al. [16] proposed an origami-based foldable sound absorber composed of six different channels made of polyethylene based on micro-perforated resonators made of photosensitive resin by 3D printing, which provided an avenue for flexible low-frequency noise control.

Present so-called adjustable sound-absorbing materials or structures can change the absorption band by altering the structural parameters. Shao et al. [19] proposed a muffler with a membrane structure and multi-Helmholtz cavities through the 3D printing of polylactic resin. The reduction frequency can be adjusted by changing the number of membranes and the width of the reduction frequency can be expanded by increasing the number of cavities. Pei et al. [20] theoretically constructed a rectangular PC with C2v symmetry and placed four steel rectangular scatters at each quarter position inside the crystal, and the final composite crystal had two different forms: the V-type and the T-type. The acoustic–structural interaction of the membrane-type acoustic metamaterial made of the hyper-elastic material of dielectric elastomer film with eccentric masses has been explored by Lu et al. [21]. This scheme was set to optimize the distribution of the eccentric masses to improve acoustic performance. A new sort of perforated sound absorber and metamaterial-based micro-perforated panels (MMPPs) were developed by Ren et al. [22]. It was found that increasing the additional mass ratio and damping the LRs, as well as using multiple local resonances, can further increase and extend absorption enhancement. However, the adjustment requires the refabrication of the new sound absorber [19,20,21,22], which limits their practical applications in noise abatement.

Meanwhile, the development of 3D printing technology makes the design and fabrication of acoustic metamaterials with a complex shape possible, especially the low-force stereolithography technique. Thus, the adjustable acoustic metamaterial of multiple parallel-connection Helmholtz resonators with tunable apertures (TA–MPCHRs) was proposed and prepared by the low-force stereolithography of photopolymer resin in this study. First of all, the effect of the tunable aperture diameter was investigated through acoustic finite element simulations with a two-dimensional rotationally symmetric model. Later, the TA–MPCHR with a total thickness of 40 mm was then designed to meet the noise reduction effects in the frequency ranges 500–800 Hz, 550–900 Hz, 600–1000 Hz and 700–1150 Hz, according to the practical application requirements. Afterwards, the actual TA–MPCHR sample was made of photopolymer resin by low-force stereolithography according to the optimal parameters. Finally, the effectiveness of the TA–MPCHR was verified based on the experimental and simulation results. At the same time, the proposed TA–MPCHR was compared with common non-adjustable MPCHRs to verify the sound absorption effect.

## 2. Materials and Methods

### 2.1. Structural Design

The present so-called adjustable sound-absorbing materials or structures can change the absorption band by altering the structural parameters, but this adjustment requires the refabrication of the new sound absorber [15,16,17,18], which limits their practical applications in noise abatement. In this work, the acoustic metamaterial of multiple parallel-connection Helmholtz resonators with tunable apertures (TA–MPCHRs) was proposed and is shown in Figure 1a. It was divided into two parts, consisting of the back chamber shown in Figure 1b and the front panel shown in Figure 1c. Meanwhile, taking the following measurements into account, the proposed TA–MPCHR contained 5 metamaterial cells, as shown in Figure 1a, and the cell G1, as shown in Figure 1d. The metamaterial cell was the basic structural unit for sound absorption with broadband, and it could be infinitely expanded in practical applications. It should be noted that, with the exception of metamaterial cell G1, the arrangements of the other 4 metamaterial cells (G2 to G5) were adjusted to maintain the required size of the TA–MPCHR sample in Figure 1a. This adjustment had no influence on sound absorption performance because the major influencing factors were the structural parameters of Helmholtz resonators instead of the distribution of their positions for a given TA–MPCHR sample.

Furthermore, each metamaterial cell contained 9 single Helmholtz resonators, and the schematic drawing and cutaway view of a single Helmholtz resonator are shown in Figure 2a and Figure 2b, respectively. Moreover, the influencing parameters of the single Helmholtz resonator on its sound absorption performance are marked in Figure 2b, which consists of the diameter of the aperture *d*, length of the aperture *l*, thickness of the front and back panel *t*_0_, thickness of the side wall *t*_1_, length of the cavity *T* and side length of the cavity *a* (section of the chamber was square). These parameters were identical for each Helmholtz resonator except *l*. The tunable aperture was realized by 4 segments of the extendable cylindrical chambers, and its schematic drawing and cutaway view are shown in Figure 2c and Figure 2d, respectively. The 4 segments of the cylindrical chambers had same thickness *t*_2_ and length *l*_0_, and each diameter was approximate to the diameter of the aperture *d* because *d* >> *t*_2_, as shown in Figure 2d. In actual applications, the distance between the neighboring metamaterial cells was their side length 3*a* + 3*t*, and that between the adjacent single Helmholtz resonators was their side length *a* + *t*.

Thus, it could be observed that the length of the aperture *l* was in the range of [*l*_0_, 4*l*_0_], and its value could be tunable by adjusting the 4 segments of the cylindrical chambers. The roots of all the tunable apertures were fixed on the front panel, as shown in Figure 1c. With the exception of the tunable aperture which was made of an aluminum alloy, the parts of the TA–MPCHR sample were prepared by the low-force stereolithography of the photopolymer resin. By this method, the acoustic metamaterial of the TA–MPCHR sample was achieved, and the related specific parameters are summarized in Table 1.

### 2.2. Theoretical Modeling

The theoretical sound absorption coefficient of the acoustic metamaterial of the TA–MPCHR can be obtained according to electro-acoustic theory [8,9,13,17], as shown in Equation (1). Here, α is the theoretical sound absorption coefficient, Z is the total acoustic impedance of the metamaterial cell, $\rho_{0}$ is the density of the air and $c_{0}$ is the sound velocity in air.
(1)$$\alpha =1-\left|\frac{Z/\rho_{0}/c_{0}-1}{Z/\rho_{0}/c_{0}+1}\right|$$

The total acoustic impedance of the metamaterial cell Z can be obtained by the parallel connection of the single resonators, as shown in Equation (2). Here, $Z_{n}$ is the acoustic impedance of the *n*th single resonator, which includes the acoustic impedance of the aperture $Z_{nm}$ and the acoustic impedance of the rear cavity $Z_{nc}$, as shown in Equation (3).
(2)$$Z=1/\sum_{n=1}^{16}\left(\frac{1}{Z_{n}}\right)$$
(3)$$Z_{n}=Z_{nm}+Z_{nc}$$

The acoustic impedance of the aperture $Z_{nm}$ can be derived by Equation (4) based on the Euler equation. Here, ω is the sound angular frequency, $l_{n}$ is length of the aperture, $\sigma_{n}$ is the perforation ratio, $B_{1}\left(\eta_{n}\sqrt{-i}\right)$ and $B_{0}\left(\eta_{n}\sqrt{-i}\right)$ are the first order and zero order Bessel functions of the first kind, respectively, $\eta_{n}$ is the perforation constant, which can be obtained by Equation (5), μ is the dynamic viscosity coefficient of the air and $d_{n}$ is the diameter of the hole.
(4)$$Z_{nm}=i\frac{\omega \rho_{0}l_{n}}{\sigma_{n}}{\left[1-\frac{2B_{1}\left(\eta_{n}\sqrt{-i}\right)}{\left(\eta_{n}\sqrt{-i}\right)\cdot B_{0}\left(\eta_{n}\sqrt{-i}\right)}\right]}^{-1}+\frac{\sqrt{2}\mu \eta_{n}}{\sigma_{n}\cdot d_{n}}+i\frac{0.85\omega \rho_{0}\cdot d_{n}}{\sigma_{n}}$$
(5)$$\eta_{n}=d_{n}\sqrt{\frac{\rho_{0}\omega }{4\mu }}$$

The acoustic impedance of the rear cavity $Z_{nc}$ can be achieved through the impedance transfer formula, as shown in Equation (6). Here, $Z_{nce}$ is the effective characteristic impedance of the air in the cavity, which can be obtained by Equation (7). $k_{nce}$ is the effective transfer constant of the air in the cavity, which can be achieved by Equation (8), and T is the thickness of the cavity.
(6)$$Z_{nc}=-iZ_{nce}\cot\left(k_{nce}T\right)$$
(7)$$Z_{nce}=\sqrt{\frac{\rho_{0e}}{C_{0e}}}$$
(8)$$k_{nce}=\omega \sqrt{\rho_{0e}C_{0e}}$$

In Equations (7) and (8), $\rho_{0e}$ and $C_{0e}$ are the effective density and effective volumetric compressibility of the air, respectively, which can be obtained by Equations (9) and (10). Here, v can be calculated by Equation (11). a and h are the side lengths of the cavity section, and it is a=h in this research. $\alpha_{x}^{}=\left(x+1/2\right)\pi /a$ and $\beta_{y}^{}=\left(y+1/2\right)\pi /h$ are the intermediate calculation coefficients, $P_{0}$ is the standard atmospheric pressure under a normal temperature, γ is the specific heat rate of the air, $v^{\prime }$ can be obtained by Equation (12), and κ and $C_{v}$ are the thermal conductivity and specific heat capacity at the condition of the constant volume, respectively.
(9)$$\rho_{0e}=\rho_{0}\frac{va_{}^{2}h_{}^{2}}{4i\omega }{\left\{\sum_{x=0}^{\infty }\sum_{y=0}^{\infty }{\left[\alpha_{x}^{2}\beta_{y}^{2}\left(\alpha_{x}^{2}+\beta_{y}^{2}+\frac{i\omega }{v}\right)\right]}^{-1}\right\}}^{-1}$$
(10)$$C_{0e}=\frac{1}{P_{0}}\left\{1-\frac{4i\omega \left(\gamma -1\right)}{v^{\prime }a_{}^{2}h_{}^{2}}{\sum_{x=0}^{\infty }\sum_{y=0}^{\infty }\left[\alpha_{x}^{2}\beta_{y}^{2}\left(\alpha_{x}^{2}+\beta_{y}^{2}+\frac{i\omega \gamma }{v^{\prime }}\right)\right]}^{-1}\right\}$$
(11)$$v=\mu /\rho_{0}$$
(12)$$v^{\prime }=\kappa /\rho_{0}/C_{v}$$

According to the constructed theoretical model for the sound absorption coefficient of the acoustic metamaterial of the TA–MPCHR in Equations (1)–(12), the theoretical sound absorption coefficients of the investigated metamaterial cells can be obtained. It was proved that the difference between the theoretical data with experimental data is larger than that between the simulation data with the experimental data [8,9,13,17]. A major reason for low prediction accuracy in the theoretical model is that there were many approximations and neglects in the modeling process, which resulted in large fluctuations in the distribution of sound absorption coefficients within the effective frequency range and the deviation of the absorption peak frequencies. Generally speaking, distributions of the sound absorption coefficients in the simulation were consistent with those in the experiments because the finite element model simulates the actual measurement process. Therefore, analysis of the sound absorption mechanism is conducted by acoustic finite element simulation instead of the theoretical model.

### 2.3. Influence of the Tunable Aperture

The influence of the tunable aperture with a variable length was preliminarily investigated through the acoustic finite element simulation with a two-dimensional rotational symmetric model, as shown in Figure 3a, which aimed to improve research efficiency by finding some guidance by adjusting the length of the different chambers in the metamaterial cell. The model was built according to the parameters in Table 1, and the square chamber with a side length of 10 mm was equivalent to a circular chamber with a diameter of 11.3 mm by keeping their areas equal. The simulation model consisted of a perfect matching layer, background acoustic field [19], embedded aperture and chamber, and it was further gridded with free triangle mesh, distribution and boundary stretching [20,21,22], as shown in Figure 3b. Through the parametric sweep of the length of the aperture l in the range of 4–20 mm with an interval of 2 mm, the theoretical sound absorption coefficients were obtained, as shown in Figure 3e. The distribution of the acoustic velocity and that of the temperature are shown in Figure 3c and Figure 3d, respectively, which could intuitively exhibit the sound absorption mechanism in a single Helmholtz resonator. It could be found that when the frequency of the incident wave was consistent with the resonance frequency of the Helmholtz resonator, the air in the rear chamber was expanded and compressed alternately, which led to the reciprocating motion of the air in the aperture with high speed, as shown in Figure 3c. This generated the thermal viscosity effect between the moving air and the wall of the rear chamber and the thermal viscosity between the moving air and the wall of the aperture. Additionally, it resulted in the conversion of sound energy to heat energy and the increase in the temperature of the air in both the chamber and the aperture [23,24,25,26], which could be judged from the distribution of the temperature in Figure 3d. It can be seen in Figure 3e that, along with the decrease in *l*, the resonance frequency shifted to the high frequency direction and the peak sound absorption coefficient rose gradually.

Therefore, the desired sound absorber based on the proposed acoustic metamaterial of the TA–MPCHR could be obtained by adjusting the length of the aperture *l* one by one according to the target frequency range. For example, a certain equipment had 4 operating modes, and the corresponding generated noises were in the range of 500–800 Hz, 550–900 Hz, 600–1000 Hz and 700–1150 Hz, respectively. For common MPCHRs, there are normally two popular methods for the effective control of the noise generated by this equipment. First was the development of the sound absorber with the broad absorption band of 500–1150 Hz, which indicated a large thickness of the absorber to use more space, and it was difficult to design such an absorber with broadband [27]. Second was the fabrication of 4 sound absorbers with different absorption bandwidths corresponding to the 4 noise spectrums and the change of various absorbers for each kind of operating modes, which meant there were high manufacturing costs and heavy replacement workload [28]. Fortunately, the TA–MPCHR can effectively solve this kind of problem. The chambers of the metamaterial cell for this sound absorber can be fixed directly, and the required quantity of metamaterial cells depend on the area of the workshop. Meanwhile, the optimal length of the apertures can be calculated in advance for the 4 operating modes, which can be tuned simply by adjusting the 4 segments of the cylindrical chambers, and the well-tuned front panel is installed to the back chamber of metamaterial cell. Using this method, various sound absorption properties were gained by the proposed acoustic metamaterial of the TA–MPCHR and the adjustment was easy to realize. The sound absorption coefficients of the TA–MPCHR sample were detected through an AWA6290T detector, referring to the national standard of GB/T 18696.2–2002 (ISO 10534–2:1998) based on the transfer function method [29,30,31]. The schematic diagram of the detection process is exhibited in Figure 4.

The actual sound absorption coefficients $\alpha_{0}$ of the TA–MPCHR with vertical incidence can be calculated by the peak value and valley value of the acoustic pressure, as shown in Equation (13).
(13)$$\alpha_{0}=\frac{I_{i}-I_{r}}{I_{i}}=1-\frac{I_{r}}{I_{i}}=1-\frac{P_{r}^{2}}{P_{i}^{2}}$$

Here, $I_{i}$, $I_{r}$, $P_{i}$ and $P_{r}$ are the incident sound intensity, the reflected sound intensity, the incident sound pressure and the reflected sound pressure, respectively. Supposing that $n=\frac{P_{\max}}{P_{\min}}=\frac{P_{i}+P_{r}}{P_{i}-P_{r}}$, Equation (13) can be converted into Equation (14).
(14)$$\alpha_{0}=1-\frac{P_{r}^{2}}{P_{i}^{2}}=\frac{4n}{{\left(n+1\right)}^{2}}$$

Acoustic pressure data in the standing wave can be achieved by the standing wave tube method. The difference of the sound level L between the peak value $L_{\max}$ and the valley value $L_{\min}$ can be calculated by Equation (15).
(15)$$L=L_{\max}-L_{\min}=20\mathrm{lg}P_{\max}/P_{0}-20\mathrm{lg}P_{\min}/P_{0}=20\mathrm{lg}P_{\max}/P_{\min}=20\mathrm{lg}n$$

According to Equations (13)–(15), the actual sound absorption coefficient $\alpha_{0}$ of MSC-AM with vertical incidence can be achieved as shown in Equation (16), and the peak value $L_{\max}$ and valley value $L_{\min}$ of the sound pressure for a certain frequency can be automatically obtained by the standing wave tube tester.
(16)$$\alpha_{0}=\frac{4\times 10^{\left(L_{\max}-L_{\min}\right)/20}}{{\left(1+10^{\left(L_{\max}-L_{\min}\right)/20}\right)}^{2}}$$

In order to improve the optimization process, there were 4 rules to obey.

(1)The length of the aperture for each Helmholtz resonator in one metamaterial cell was labelled as *l*_i_ (i = 1,2, …,9), and *l*_i_ decreased along with an increase in the serial number of the Helmholtz resonator in the metamaterial cell, which meant that *l*_1_ > *l*_2_ > *l*_3_ > *l*_4_ > *l*_5_ > *l*_6_ > *l*_7_ > *l*_8_ > *l*_9_.(2)The target frequency range [*f*_min_, *f*_max_] was divided into 9 sections, and *l*_1_ was responsible for absorbing the noise with the lowest section and *l*_9_ corresponded to the absorption of the noise with the highest section.(3)The optimization objective was a maximum average sound absorption coefficient *α*(*f*) in the target frequency range.(4)The optimization would be stopped if the conditions of the average sound absorption coefficient were not improved, or if the improvement extent was smaller than 1% in the latest 20 measurements.

The gained optimal parameters of the TA–MPCHR sample for the given target are summarized in Table 2, and the corresponding sound absorption performances are exhibited in Figure 5a–d. It could be observed that excellent sound absorption performances were obtained for the various frequency ranges with a limited total thickness of 40 mm (*T* + 2 × *t*_0_).

### 2.4. Analysis of the Sound Absorption Mechanism of TA–MPCHR

In order to study the sound absorption mechanism of the TA–MPCHR, the three-dimensional acoustic finite element simulation model was built, as shown in Figure 6a. It is further gridded in Figure 6b, which is similar with the two-dimensional rotational symmetric model in Figure 3b. Additionally, the theoretical sound absorption coefficients were obtained as shown in Figure 5a–d. It can be seen that the theoretical values were consistent with the actual experimental data, and deviations were generated by idealized conditions in the simulation process. The arrangement of 5 metamaterial cells and that of 9 Helmholtz resonators in each metamaterial cell are exhibited in Figure 6c and are consistent those shown in Figure 1a,d. The parameters for the gridded mesh consisted of a maximum cell size of 3.7 mm, a minimum cell size of 0.037 mm, a maximum unit growth rate of 1.3, a curvature factor of 0.2, and the resolution of the narrow region which was 0.95. In order to increase the simulation accuracy, the boundary layers were refined with the stretching factor 1.2 and the regulatory factor of their thicknesses 1. Meanwhile, the plane wave with the amplitude 1 was utilized in the background acoustic field as the incident sound wave with normal incidence, and the perfect matching layer was used to simulate the perfect absorption of the sound waves in the process of propagation away from the sound source. The diameter of the front panel *D* was equal to the diameter of the perfect matching layer and that of the acoustic background field, and it was selected as 100 mm in this study, which was consistent with the actual diameter of the proposed acoustic metamaterial of the TA–MPCHR sample. The direct linear solver of PARDISO was selected as a steady-state solver in the finite element simulation procedure which could utilize shared-memory parallel processing to improve the computational efficiency.

For the operating mode of the equipment with a frequency range of 600–1000 Hz, the distributions of temperature for the frequency point with an interval of 50 Hz are shown in Figure 7a–i. It can be observed that the sound absorption at the resonance frequency was mainly determined by the corresponding Helmholtz resonator and the sound absorption at the other frequency point was achieved using the coupling effect of all Helmholtz resonators, which was consistent with the normal principle for common MPCHRs [32]. By contrast, for the sound absorbers of common non-adjustable MPCHRs with various total thicknesses of 40–80 mm by intervals of 10 mm, they were optimized in a finite element simulation model to achieve a broad absorption band of 500–1150 Hz with a maximal average sound absorption coefficient. With the exception of the cavity T and that of the apertures li, the parameters were the same as those of the TA–MPCHR in Table 1.

### 2.5. Result and Discussion

The optimal parameters for common non-adjustable MPCHRs with various total thicknesses are summarized in Table 3, and the corresponding distributions of the theoretical sound absorption coefficients are shown in Figure 8. It should be noted that the length of the aperture li might be smaller than the thickness of the front panel t0 to realize the effective sound absorption in a high frequency region, which indicated that the front panel should be thinned for the corresponding Helmholtz resonators. It could be found that, in order to gain an average sound absorption coefficient of 0.9202, which was the minimum value among 4 average sound absorption coefficients Table 2, the total thickness of common non-adjustable MPCHRs should be close to 80 mm, which was almost twice the total thickness of the TA–MPCHR (40 mm). These results further proved that the proposed acoustic metamaterial of the TA–MPCHR could significantly reduce the occupied space and improve sound absorption efficiency.

Meanwhile, the specific parameters of the TA–MPCHR sample in Table 1 can be modified for various applications. The resonance frequency *f*_0_ of a single Helmholtz resonator was derived by Equation (17) [33] and the meanings of the parameters were the same as those in Figure 2b. Here, *c*_0_ was the sound velocity in air.
(17)$$f_{0}=\frac{c_{0}d}{8a\sqrt{\pi }}\sqrt{\frac{1}{lT}}$$

It can be judged from Equation (1) that the *f*_0_ would shift to a low frequency direction by decreasing the *d* or increasing the *a*, *l* or *T*. However, the side length of the cavity *a* and the diameter of the aperture *d* were difficult to adjust for an already well-produced MPCHR. Meanwhile, the adjustment of the length of the cavity T for each Helmholtz resonator was indicated by an inhomogeneous back panel, which resulted in the waste of space for all the Helmholtz resonators, except for the longest one [34]. Therefore, the length of the aperture *l* was the most suitable option to tune the sound absorption performance for a well-produced MPCHR. Moreover, for certain applications of noise reduction, specific parameters of the proposed TA–MPCHR were decided by the number of required modes and the target frequency range for each mode, which could be preliminarily derived according to the causal nature of acoustic response [35]. Meanwhile, the proposed TA–MPCHR was a continuous expansion of the uniform metamaterial cell, and the optimization of two-dimensional expansion and the arrangement of various metamaterial cell was also an efficient method [36,37,38] to extend sound absorption bandwidth, which represents the future research focus, i.e., to further improve sound absorption performance and promote actual applications.

## 3. Conclusions

In summary, the acoustic metamaterial of the TA–MPCHR was proposed in this study, and it was experimentally demonstrated to achieve a broad low-frequency sound absorption band. For the four various operation modes of certain equipment, average sound absorption coefficients of 0.9203, 0.9202, 0.9436 and 0.9561 were obtained with a limited total thickness of 40 mm, corresponding to the target frequency ranges of 500–800 Hz, 550–900 Hz, 600–1000 Hz and 700–1150 Hz. Relative to the common non-adjustable MPCHR, the proposed TA–MPCHR could save almost half of the occupied space, which is favorable in the promotion of its practical applications.

## Figures and Tables

**Figure 1 polymers-14-05434-f001:**
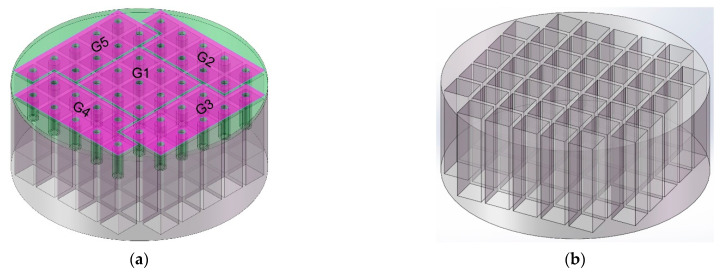
The proposed acoustic metamaterial of TA–MPCHR. (**a**) The overall structure of the TA–MPCHR. (**b**) The back chamber. (**c**) The front panel. (**d**) The metamaterial cell.

**Figure 2 polymers-14-05434-f002:**
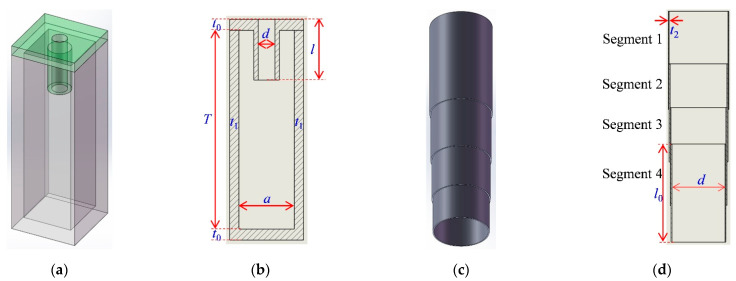
The detailed view of TA–MPCHR. (**a**) The single Helmholtz resonator. (**b**) The cutaway view of single Helmholtz resonator. (**c**) The tunable aperture. (**d**) The cutaway view of tunable aperture.

**Figure 3 polymers-14-05434-f003:**
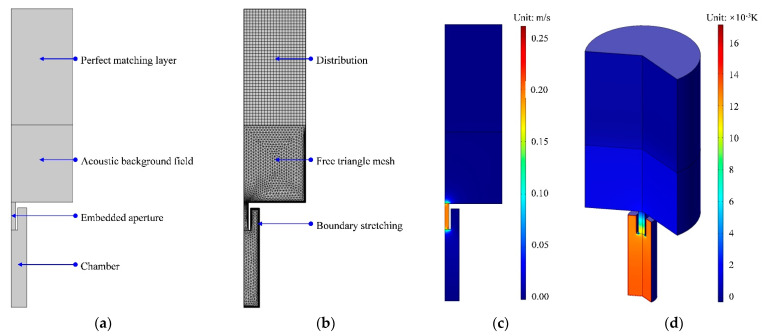
The acoustic finite element simulation of sound absorption coefficient of single Helmholtz resonator. (**a**) The two-dimensional rotational symmetric model. (**b**) The gridding model. (**c**) The distribution of acoustic velocity. (**d**) The distribution of temperature. (**e**) The distribution of sound absorption coefficients with different length of the aperture *l*.

**Figure 4 polymers-14-05434-f004:**
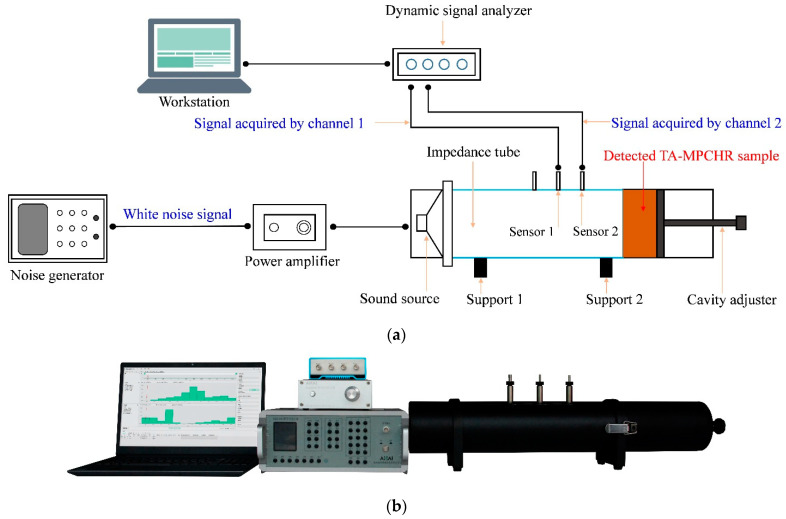
AWA6290T detector. (**a**) Schematic diagram; (**b**) Actual image.

**Figure 5 polymers-14-05434-f005:**
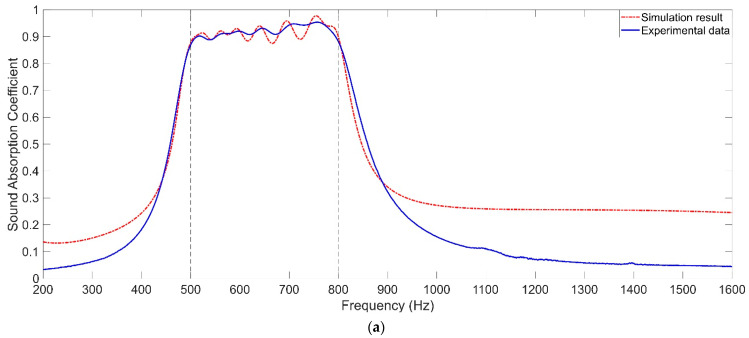
Detection of the TA–MPCHR sample. (**a**) Absorption property for 500–800 Hz. (**b**) Absorption property for 550–900 Hz. (**c**) Absorption property for 600–1000 Hz. (**d**) Absorption property for 700–1150 Hz.

**Figure 6 polymers-14-05434-f006:**
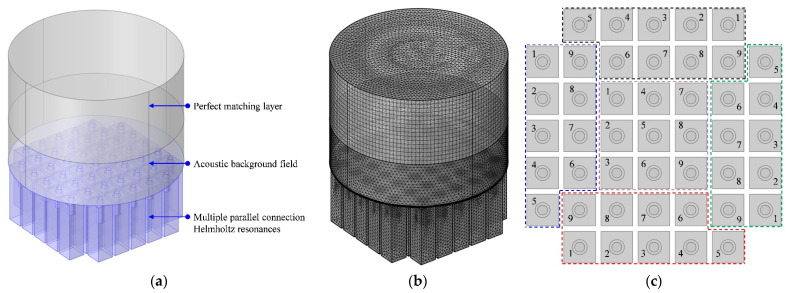
The finite element simulation of TA–MPCHR sample. (**a**) Three-dimensional acoustic finite element simulation model. (**b**) The gridding model. (**c**) The arrangement of metamaterial cells and Helmholtz resonators.

**Figure 7 polymers-14-05434-f007:**
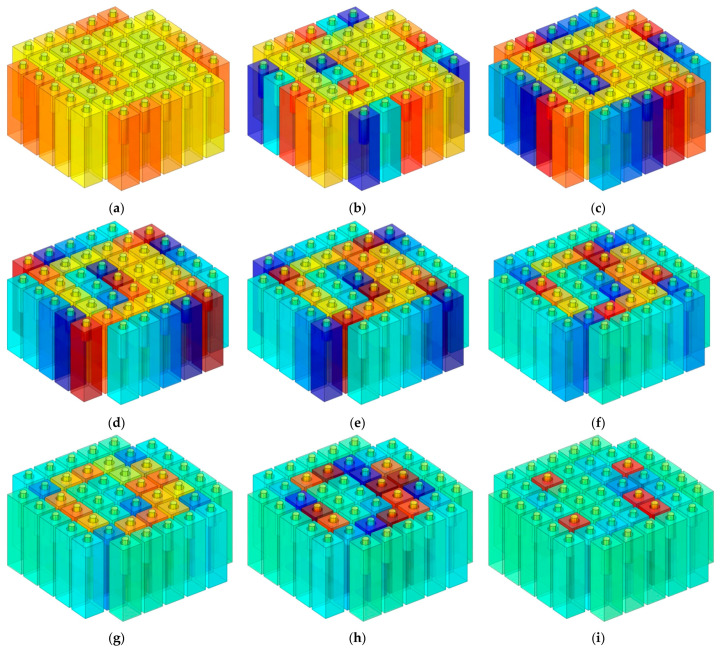
The sound absorption mechanism of TA–MPCHR sample. (**a**) For 600 Hz. (**b**) For 650 Hz. (**c**) For 700 Hz. (**d**) For 750 Hz. (**e**) For 800 Hz. (**f**) For 850 Hz. (**g**) For 900 Hz. (**h**) For 950 Hz. (**i**) For 1000 Hz.

**Figure 8 polymers-14-05434-f008:**
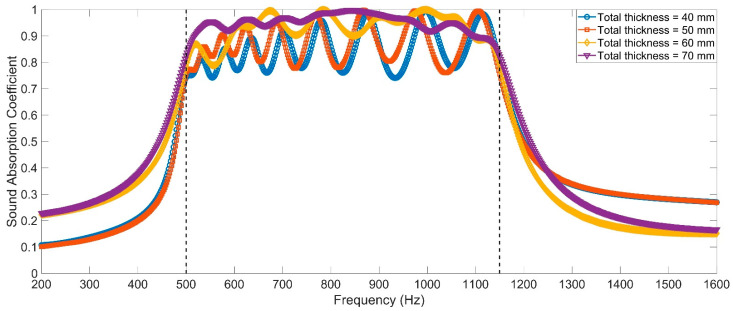
Distributions of theoretical sound absorption coefficients of the optimized common non-adjustable MPCHR.

**Table 1 polymers-14-05434-t001:** Specific parameters of the proposed acoustic metamaterial of TA–MPCHR sample (mm).

Parameters	Cavity		Front/Back Panel	Side Wall	Aperture		
	Side Length *a*	Length *T*	Thickness *t*_0_	Thickness *t*_1_	Diameter *d*	Length *l*_0_	Thickness *t_2_*
**Values**	10	36	2	1.8	3.16	6	0.02

**Table 2 polymers-14-05434-t002:** The summarized optimal parameters of TA–MPCHR sample.

Target Frequency Range	Length of the Aperture *l*_i_ (mm)									*α*(*f*)		
	*l* _1_	*l* _2_	*l* _3_	*l* _4_	*l* _5_	*l* _6_	*l* _7_	*l* _8_	*l* _9_	Average	Maximum	Minimum
**500–800 Hz**	25.7	23.9	21.7	18.6	16.1	13.4	10.8	8.6	7.4	0.9203	0.9539	0.8715
**550–900 Hz**	21.2	19.2	17.0	14.7	12.3	10.0	7.9	6.4	5.3	0.9202	0.9552	0.8348
**600–1000 Hz**	17.2	15.6	13.6	11.5	9.6	7.8	6.1	4.5	3.6	0.9436	0.9832	0.8474
**700–1150 Hz**	11.2	10.3	9.0	7.5	6.1	4.7	3.4	2.2	2.0	0.9561	0.9817	0.8449

**Table 3 polymers-14-05434-t003:** The summarized optimal parameters of the optimized common non-adjustable MPCHR.

Total Thickness (mm)	Diameter of the Aperture (mm)	Length of the Aperture *l*_i_ (mm)									*α*(*f*)		
		*l* _1_	*l* _2_	*l* _3_	*l* _4_	*l* _5_	*l* _6_	*l* _7_	*l* _8_	*l* _9_	Average	Maximum	Minimum
40	3.16	25.3	21.7	17.6	14.1	10.9	8.2	5.8	3.7	2.2	0.8421	0.9961	0.7412
50	3.16	20.9	17.5	14.4	11.4	8.5	5.9	3.7	2	0.6	0.8693	0.9987	0.7542
60	4.28	28.2	24.1	20	16	12.1	8.2	4.7	1.6	0.1	0.8926	0.9546	0.7509
70	5.44	35.8	29.7	23.8	18.3	13.3	8.7	4.9	1.9	0.1	0.9119	0.9541	0.7885
80	6.62	38.7	32.6	26.6	20.7	14.9	9.7	5.4	1.9	0.1	0.9229	0.9753	0.8496

## Data Availability

The processed data required to reproduce these findings are already included in the article. The raw simulation data and experimental data required to reproduce these findings are also available upon the request by contact with the corresponding author.

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
