# Peer review of "Adjustable Sound Absorber of Multiple Parallel-Connection Helmholtz Resonators with Tunable Apertures Prepared by Low-Force Stereolithography of Photopolymer Resin"

_polymers, 2022, doi:10.3390/polym14245434_

Round 1

Reviewer 1 Report

The article is interesting and relevant. modification and application of resonators is a widely studied problem. The paper should provide data on the distances between the resonators so that the study can be replicated.

It would also be good to provide sample photos. Please explain what program or on what basis the theoretical efficiency of resonators was predicted, so that other authors can also use it.

Please provide dimensions in Figure 6. provided by the A–MPCHR sample.

Author Response

Response to reviewer 1

General Comment: The article is interesting and relevant. Modification and application of resonators is a widely studied problem.

Response:

Thank you very much for your kind review to our manuscript and the meaningful comment to our research. We have revised the manuscript carefully according to your and other reviewers’ comments. The responses to your comments are as follows.

  1. The paper should provide data on the distances between the resonators so that the study can be replicated.

Response:

Thank you very much for your kind suggestion. As shown in Figure 1, each metamaterial cell contained 9 single Helmholtz resonators, which was the basic structural unit for sound absorption with broadband and it could be infinite expanded in practical applications. Thus, the distance between the neighboring metamaterial cells was their side length 3a+3t, and the distance between the adjacent single Helmholtz resonators was their side length a+t, which could be derived from the marks in the Figures 1 and 2. In order to make the paper more readable and understandable, the distances between the resonators were added in the revised manuscript, and this modification were highlighted in yellow.

  1. It would also be good to provide sample photos. Please explain what program or on what basis the theoretical efficiency of resonators was predicted, so that other authors can also use it.

Response:

Thank you very much for your kind suggestion. As a novel developed sound absorber, the proposed TA–MPCHR had been submitted the application for a patent and it is under the substantive examination by the patent examiners at present. Thus, it is not convenient for us to supply picture of the fabricated sample. Instead, we provided the schematic diagram of the TA–MPCHR structure in the Figure 1 and presented its working principle in the main text, which was comment treatment in the similar literatures.

Meanwhile, the theoretical efficiency of the proposed TA–MPCHR was predicted through the acoustic finite element simulation model in the Figure 6. The detailed information was added in the revised manuscript, such as structural parameters and mesh parameters of the finite element model, and these modifications were highlighted in yellow.

  1. Please provide dimensions in Figure 6. provided by the A–MPCHR sample.

Response:

Thank you very much for your kind suggestion. The detailed information of acoustic finite element model in Figure 6 was added in the revised manuscript, which aimed to make the paper more reasonable and referable, and these modifications were highlighted in yellow.

Reviewer 2 Report

The manuscript addresses the design of an adjustable sound absorber based on multiple parallel Helmholtz resonators.

The introduction gives a good overview of the current state of the art and the goals of the work, and the results are presented and explained in a detailed an accurate way.

Some aspects should be addressed prior to the publication of this manuscript:

1.      The mathematical foundations of the work are missing in the manuscript. What model of losses is used? What is the mathematical description of the used optimization process? What are the equations for calculating the sound absorption coefficient given in the ISO standard? The authors should address the mathematical formalities prior to the analysis of the sound absorption mechanism and presentation of the results.

2.      Could the authors provide more details about the numerical model? What are the criteria for the mesh density?

3.     Somre relevant references are missing:

               * https://asa.scitation.org/doi/full/10.1121/1.4950708

               * https://www.ncbi.nlm.nih.gov/pmc/articles/PMC4726070/

               * https://www.mdpi.com/2076-3417/10/5/1690

Author Response

Response to reviewer 2

General Comment: The manuscript addresses the design of an adjustable sound absorber based on multiple parallel Helmholtz resonators.

The introduction gives a good overview of the current state of the art and the goals of the work, and the results are presented and explained in a detailed an accurate way.

Some aspects should be addressed prior to the publication of this manuscript:

Response:

Thank you very much for your kind review to our manuscript and helpful assessment to our research. We have revised the manuscript carefully according to your and other reviewers’ comments. The responses to your comments are as follows.

  1. The mathematical foundations of the work are missing in the manuscript. What model of losses is used? What is the mathematical description of the used optimization process? What are the equations for calculating the sound absorption coefficient given in the ISO standard? The authors should address the mathematical formalities prior to the analysis of the sound absorption mechanism and presentation of the results.

Response:

Thank you very much for your significant suggestion.

Theoretical sound absorption coefficient of the acoustic metamaterial of TA–MPCHR can be obtained according to Electro-Acoustic Theory, and the theoretical modeling process is added in the revised manuscript, as shown in the added section 2.2 theoretical modeling.

Sound absorption coefficients of the TA–MPCHR were detected through an AWA6290T detector referring to national standard of GB/T 18696.2–2002 (ISO 10534–2:1998) based on the transfer function method, and the equations for calculating the sound absorption coefficient was added in the revised manuscript, which were highlighted in yellow.

As mentioned in the manuscript, the parameter optimization was realized by the actual experiments instead of theoretical model or finite element simulation, and the optimization process was summarized into 4 rules. Therefore, there was no mathematical description of the used optimization process.

  1. Could the authors provide more details about the numerical model? What are the criteria for the mesh density?

Response:

Thank you very much for your significant suggestion. The detailed information of acoustic finite element model in Figure 6 was added in the revised manuscript, such as structural parameters and mesh parameters of the finite element model, which aimed to make the paper more reasonable and referable, and these modifications were highlighted in yellow.

  1. Some relevant references are missing:

https://asa.scitation.org/doi/full/10.1121/1.4950708

https://www.ncbi.nlm.nih.gov/pmc/articles/PMC4726070/

https://www.mdpi.com/2076-3417/10/5/1690

Response:

Thank you very much for your kind suggestion. The recommended articles are added in the references in the revised manuscript. Meanwhile, the references are adjusted and modified according to your and other reviewers’ comments in the revised manuscript.

Round 2

Reviewer 2 Report

Fisrt of all, I want to thank the authors for considering the comments and applying the modifications to the manuscript.

All my questions and concerns have been addressed and solved, so I don't have any further comments.